

# Magnetism of magic-angle twisted bilayer graphene

**Javad Vahedi[1], Robert Peters[2], Ahmed Missaoui[3],
Andreas Honecker[3]\* and Guy Trambly de Laissardière[3]**

**1** Technische Universität Braunschweig, Institut für Mathematische Physik,
Mendelssohnstraße 3, 38106 Braunschweig, Germany
**2** Department of Physics, Kyoto University, Kyoto 606-8502, Japan
**3** Laboratoire de Physique Théorique et Modélisation, CNRS UMR 8089,
CY Cergy Paris Université, 95302 Cergy-Pontoise Cedex, France

\* andreas.honecker@cyu.fr

## Abstract

We investigate magnetic instabilities in charge-neutral twisted bilayer graphene close to so-called "magic angles" using a combination of real-space Hartree-Fock and dynamical mean-field theories. In view of the large size of the unit cell close to magic angles, we examine a previously proposed rescaling that permits to mimic the same underlying flat minibands at larger twist angles. We find that localized magnetic states emerge for values of the Coulomb interaction $U$ that are significantly smaller than what would be required to render an isolated layer antiferromagnetic. However, this effect is overestimated in the rescaled system, hinting at a complex interplay of flatness of the minibands close to the Fermi level and the spatial extent of the corresponding localized states. Our findings shed new light on perspectives for experimental realization of magnetic states in charge-neutral twisted bilayer graphene.



# 1 Introduction

Since the experimental discovery of graphene [1], two-dimensional materials have been at the focus of intensive research in condensed-matter physics, among others because they bear great promise for technological applications, see, e.g., Refs. [2, 3]. With respect to spintronics applications [4], it could nevertheless be a disadvantage that bulk graphene is non-magnetic and one needs to resort to the enhanced density of states at the Fermi level close to defects or zigzag borders in order to drive magnetic instabilities (see Ref. [5] and references therein). Recently, a twist appeared in the field when superconducting and correlated insulating states were discovered in experiments on bilayer graphene where one layer is rotated with respect to the other by a so-called "magic" angle [6,7], see Fig. 1(a) for an illustration of such a "twisted" honeycomb bilayer, Ref. [8] for a summary of some recent developments, and Refs. [9–14] for examples of resulting theoretical efforts. Even if the nature of the correlated insulating state in these systems remains under debate (see, e.g., Refs. [15–25]), it is reminiscent of the textbook antiferromagnetic insulator that appears in the Hubbard model for strong on-site Coulomb interaction $U$ [26]. Indeed, the defining feature of the magic angles [27–31] is the emergence of flat minibands around the Fermi level such that the relative importance of intrinsic interactions in graphene is enhanced. It has been demonstrated experimentally that ferromagnetism emerges when a suitable number of electrons is doped into these flat bands [32], a fact that might actually be a manifestation of the general phenomenon of flat-band ferromagnetism in the Hubbard model for suitable filling fractions [33].

Here we reexamine the one-band Hubbard model for twisted bilayer graphene (TBG) and demonstrate that magnetism occurs also in the charge-neutral (half-filled) system at low values of the on-site Coulomb interaction $U$, thus placing magnetic states, including an antiferromagnetic one, among the competitors for the instabilities in charge-neutral magic-angle twisted bilayer graphene.

We start from the tight-binding model of Refs. [28, 31]. The resulting non-interacting band structure at the first magic angle $\theta = 1.08°$ is shown by the full blue line in Fig. 1(b) and the corresponding total density of states (DOS) in Fig. 1(c). The four flat minibands and the related strong enhancement of the DOS at $E_F$ are evident. On top of that, we add Coulomb interactions between the electrons in terms of a local on-site Hubbard interaction $U$. The resulting magnetic instabilities are then investigated by a combination of real-space static mean-field theory (MFT) [5] and dynamical mean-field theory (DMFT) [34–36]. As an alternative to MFT, one could determine the instabilities of the paramagnetic state with a random-phase approximation (RPA) analysis [37], and we present results from such an RPA analysis in appendix A.

# 2 Geometry of twisted bilayer graphene (TBG)

Let us start by explaining the geometry of TBG in more detail. A single layer of graphene consists of carbon atoms arranged in a honeycomb lattice such that the unit cell includes two sites. We then construct a periodic commensurate bilayer structure parameterized by two integers $m$, $n$ using the method of Refs. [27, 28, 31, 38, 39]. $m$ and $n$ are coordinates with respect to the lattice vectors of a single graphene layer $\boldsymbol{a}_{1,2} = a(\sqrt{3}, \pm 1)/2$. The rotation angle for such a commensurate structure (moiré pattern) is then given by

$$\cos\theta = \frac{n^2 + m^2 + 4mn}{2(n^2 + m^2 + mn)}, \tag{1}$$

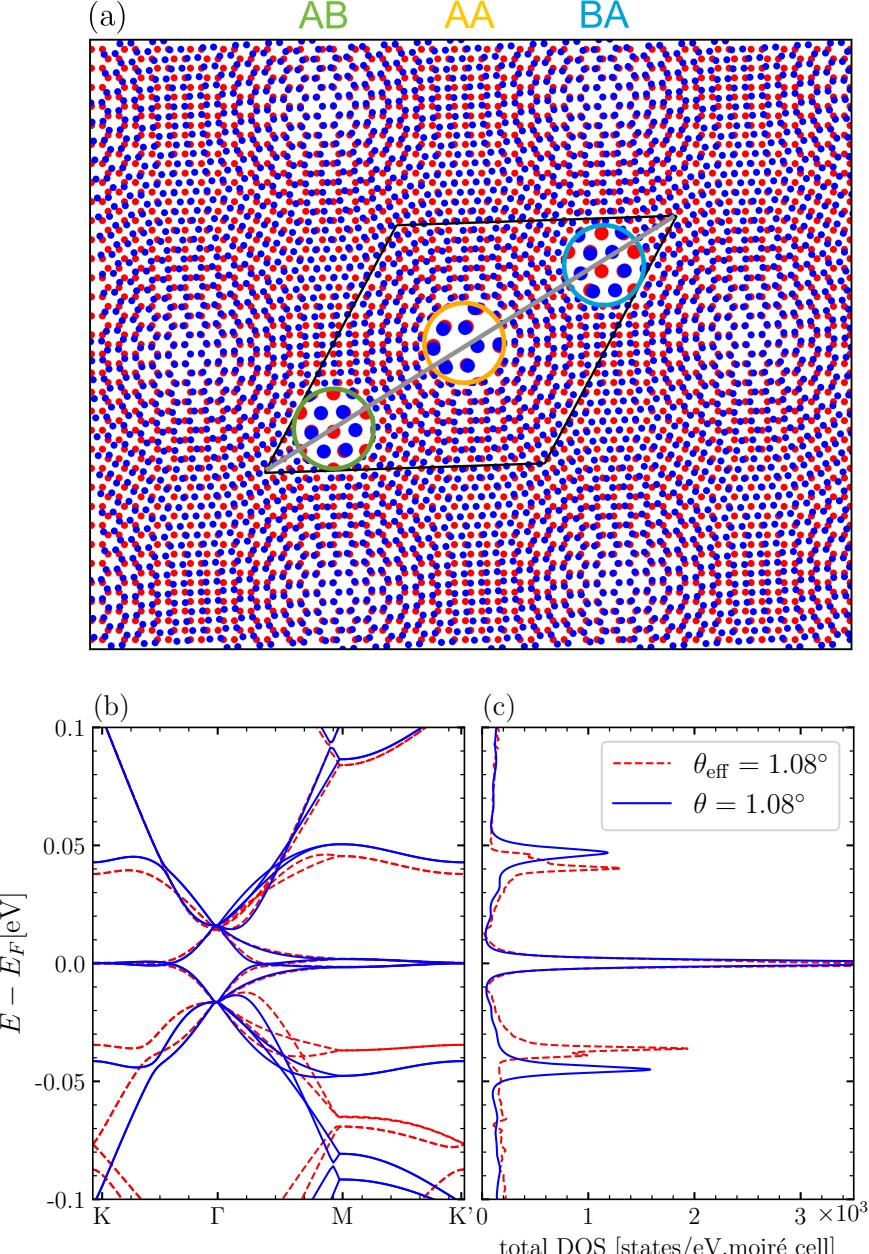

Figure 1: (a) Moiré pattern for a twist angle $\theta = 3.89°$, $[(n, m) = (8, 9)]$ with the identification of magnified regions with AB, AA, and BA stacking. (b) Band structure calculated for a system with $\theta = 1.08°$, $[(n, m) = (30, 31)]$ and $\theta_{\text{eff}} = 1.08°$, $[(n, m) = (8, 9)]$, (c) total density of states (DOS) corresponding to panel (b). The almost flat minibands at zero energy and corresponding large DOS peaks exhibit good agreement between the rescaled and non-scaled systems.

and the fundamental vectors of the TBG superlattice are $\boldsymbol{t}_1 = n\boldsymbol{a}_1 + m\boldsymbol{a}_2$ and $\boldsymbol{t}_2 = -m\boldsymbol{a}_1 + (m + n)\boldsymbol{a}_2$. The number of atoms in the moiré cell is given by

$$N_c = 4(n^2 + m^2 + mn). \tag{2}$$

Figure 1(a) shows the resulting moiré pattern for $(n, m) = (8, 9)$ corresponding to a twist angle $\theta = 3.89°$ and $N_c = 868$ atoms in the moiré cell.

## 3  Model Hamiltonian

We start from the tight-binding model for the $p_z$ orbitals of the carbon atoms in charge-neutral TBG: $\hat{H} = \hat{H}_0 + \hat{H}_{\text{int}}$, where $\hat{H}_0$ is the single-electron Hamiltonian and $\hat{H}_{\text{int}}$ is the electron-electron interaction. This leads to the one-band Hubbard model

$$\hat{H} = \sum_{i,j,\sigma} t(\boldsymbol{r}_i; \boldsymbol{r}_j)\, \hat{d}^{\dagger}_{i\sigma} \hat{d}_{j\sigma} + U \sum_i \left( \hat{n}_{i\uparrow} - \frac{1}{2} \right) \left( \hat{n}_{i\downarrow} - \frac{1}{2} \right), \tag{3}$$

where $\hat{d}^{\dagger}_{i\sigma}$ and $\hat{d}_{i\sigma}$ are the creation and annihilation operators of an electron with spin projection $\sigma = \{\uparrow, \downarrow\}$ at site $i$ and $\hat{n}_i = \sum_{\sigma} \hat{d}^{\dagger}_{i\sigma} \hat{d}_{i\sigma}$ is the total electron density at site $i$. The hopping parameters $t(\boldsymbol{r}_i; \boldsymbol{r}_j)$ between two $p_z$ orbitals located at $\boldsymbol{r}_i$ and $\boldsymbol{r}_j$ are given in Refs. [28, 31]. The second term in Eq. (3) describes the on-site Coulomb repulsion. The resulting non-interacting band structure ($U = 0$) at the first magic angle $\theta = 1.08°$ is shown by the full blue line in Fig. 1(b). This case corresponds to $(n, m) = (30, 31)$ and thus to a moiré cell with $N_c = 11164$ sites. Dealing with such big unit cells will be challenging even for a one-band model and even within mean-field theory (MFT) and thus we will explore an idea of Ref. [40] to reduce the numerical effort.

The precise non-interacting band structure depends not only on the geometry, but evidently also on the hopping parameters $t(\boldsymbol{r}_i; \boldsymbol{r}_j)$, and in particular the ratio between intra- and interlayer hopping. Let $\theta$ and $\theta'$ be the angles corresponding to two commensurate moiré structures and

$$\Lambda = \frac{\sin\frac{\theta'}{2}}{\sin\frac{\theta}{2}}. \tag{4}$$

Then the rescaling $t'_0 = \Lambda t_0$ of the nearest-neighbor intralayer hopping while keeping the interlayer hopping unchanged maps the low-energy band structure from the unprimed to the primed geometry [40]. The panels (b) and (c) of Fig. 1 illustrate this mapping for the first magic angle from $\theta = 3.89°$ to $\theta_{\text{eff}} \equiv \theta' = 1.08°$. Indeed, the dashed red line reproduces both the low-energy band structure and the density of states well at the expense of reducing the nearest-neighbor intralayer hopping from the physical value $t_0 = 2.7\,\text{eV}$ [5, 41] to $t'_0 \approx 0.75\,\text{eV}$, *i.e.*, modifying the high-energy physics. With different rescaling factors, *i.e.*, $t'_0 \approx 0.90\,\text{eV}$ and $1.02\,\text{eV}$, we can also model the angles $\theta = 1.30°$ and $1.47°$ in the $(n, m) = (25, 26)$ and $(22, 23)$ systems, respectively by the same effective $(n, m) = (8, 9)$ system.

Ref. [40] suggested that the on-site Coulomb interaction should scale in the same way as the intralayer hopping parameters, $U' = \Lambda U$ although this is less evident than the rescaling of the hopping parameters, as we will also see in the results to be presented below.

In the following section 4 we will first explore this rescaling trick in order to perform a detailed study using the case $(n, m) = (8, 9)$ ($N_c = 868$). In section 5 we will then check for some representative cases to what extent the conclusions do indeed apply to the unscaled system, including the first magic angle, *i.e.*, $(n, m) = (30, 31)$ ($N_c = 11164$).

## 4  Rescaled system

In this section, we investigate the Hubbard model (3) for twisted bilayer graphene (TBG) using rescaled interlayer hopping parameters, as outlined in the previous section. We will start with a systematic study using static MFT and then use a more sophisticated dynamical mean-field theory (DMFT) to argue that the findings of the simple MFT are qualitatively correct even if there is a quantitative renormalization of the values of the on-site Coulomb interaction $U$.

## 4.1 Static mean-field theory (MFT)

Static MFT is a well-established method to investigate the magnetism in graphene (see, e.g., chapter 3.1 of [5] and Refs. [35,36,40,42–44]) such that here we summarize only the essential features. It amounts to the Hartree-Fock approximation of the interaction term in Eq. (3),

$$U n_{i\uparrow} n_{i\downarrow} \approx U \left( \langle n_{i\uparrow} \rangle n_{i\downarrow} + \langle n_{i\downarrow} \rangle n_{i\uparrow} - \langle n_{i\uparrow} \rangle \langle n_{i\downarrow} \rangle \right), \tag{5}$$

where $\langle n_{i\sigma} \rangle$ is the average electron occupation number with spin $\sigma$ at site $i$. Note that the approximation (5) decouples the operators for the two spin sectors and thus gives rises to a quadratic Hamiltonian in each of them where the other spin sector enters only via the site-dependent mean fields $\langle n_{i\sigma} \rangle$ that have to be determined self-consistently. We focus on charge-neutral TBG that has exactly one electron per site, i.e., we work with the half-filled Hubbard model. A self-consistent solution is found iteratively, where in each step $N_c \times N_c$ matrices need to be diagonalized and an integral over the moiré Brillouin zone has to be calculated, that we approximate by a uniform grid of $k$ points. We iterate this procedure until the maximum change of a density is below $10^{-6}$. Given the necessity to diagonalize a large number of moderately-sized matrices, even this elementary MFT approach becomes CPU-time intensive in the present situation. Some checks indicate that a $k$-grid of at least $9 \times 9$ points is required to eliminate artifacts of this discretization while more points do not change the conclusions. We therefore show results below that have been obtained for $9 \times 9$ $k$ points.

The RPA analysis that we present in appendix A reveals different competing magnetic instabilities at different values of $q$ for the present model. There is a periodic solution with an antiferromagnetic internal structure. The dominant instabilities are actually found at $q \neq 0$, i.e., they should have a larger unit cell than the twisted bilayer lattice, and they have a ferromagnetic structure inside a moiré cell. Motivated by the fact that the Hubbard model on a single honeycomb layer becomes antiferromagnetic at large $U$ [45], we focus here on the antiferromagnetic mean-field solution. The RPA analysis of appendix A predicts a critical value $U_c \approx 0.23\, t_0'$ for the antiferromagnetic state of the twisted bilayer system with $\theta_{\text{eff}} = 1.08°$, an order of magnitude below the critical value of a single layer, that for nearest-neighbor hopping is known to be $U_c^{\text{MFT}}/t \approx 2.23$ [45].

Figure 2 shows MFT results for the total magnetization per moiré cell and its maximum value, defined as

$$M_{\text{total}} = \sum_i^{N_c} |m_z(\vec{r}_i)|, \qquad m_z(\vec{r}_i) = \frac{\langle n_{i\uparrow} \rangle - \langle n_{i\downarrow} \rangle}{2}, \tag{6}$$

$$M_{\text{max}} = \max\{|m_z(\vec{r}_1)|, \cdots, |m_z(\vec{r}_{N_c})|\}, \tag{7}$$

respectively. We first focus on the first magic angle $\theta_{\text{eff}} = 1.08°$ (red data in Fig. 2). Here, we find a small albeit finite magnetization for values of $U/t$ as low as

$$U_{c1,\text{MFT}}^{1.08°}/t_0' \approx 0.32. \tag{8}$$

We note that convergence is delicate close to $U_{c1,\text{MFT}}$ and sensitive to the chosen $k$ grid. The result (8) should thus be considered as an upper bound. Thus, we conclude that this value is consistent with the prediction of the RPA analysis of appendix A. Given that the magnetization for these small values is due to the four flat minibands and that there is a low number of associated states (4 per moiré cell), the total magnetization (6) for small values of $U$ is small and thus seen more clearly in the inset of Fig. 2(a) than in the main panel. Indeed, for $U/t_0' \lesssim 1.5$, the total magnetization per moiré cell remains below $2 = 4 \cdot 1/2$, consistent with it coming mainly from the four flat minibands.

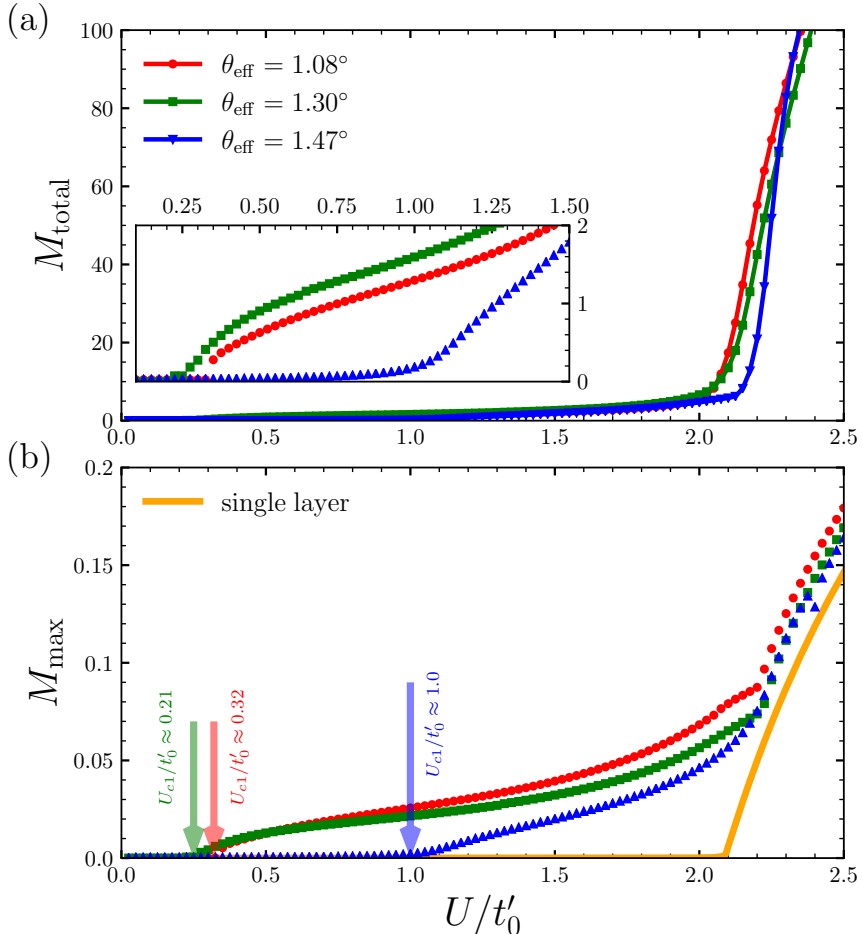

Figure 2: MFT results for the magnetization of the rescaled twisted bilayer system as a function of on-site Coulomb interaction $U/t'_0$. Panels (a) and (b) show the total magnetization per effective $N_c = 868$ moiré cell and its maximum, respectively. For comparison, results for a single graphene layer with the same intralayer hopping parameters are also shown in panel (b). In panel (b), red, green, and blue arrows mark the critical points for $\theta_{\mathrm{eff}} = 1.08°$, $\theta_{\mathrm{eff}} = 1.30°$, and $\theta_{\mathrm{eff}} = 1.47°$, respectively.

An alternative perspective is given by the maximum magnetization (7) that is shown in Fig. 2(b). Here, one can firstly observe the onset of magnetization around $U_{c1}$ more clearly than in the main panel of Fig. 2(a). For comparison, the main panel of Fig. 2(b) also includes the result for a single layer with the same intralayer hoppings as in the twisted bilayer system. One observes firstly that additional long-range hoppings within each layer reduce the critical value slightly to $U_c^{\mathrm{MFT}}/t \approx 2.09$ as compared to the nearest-neighbor result $U_c^{\mathrm{MFT}}/t \approx 2.23$ [45]. In the region $U/t'_0 \gtrsim 2$, the magnetization of the bilayer system is slightly enhanced with respect to the single-layer case, as might be expected thanks to the additional intralayer couplings. However, the transition to full magnetization necessarily involves AB and BA stacking regions (see Fig. 1(a)) that are geometrically frustrated. Consequently, one expects a complex magnetic state in this transition region. A full analysis of the transition to a fully magnetized system is beyond the scope of the present work, but we note that convergence is delicate also in this second transition region, as exemplified by the outlier at $U/t'_0 = 2.4$ in the $\theta_{\mathrm{eff}} = 1.47°$ data.

The most important finding in the present context is that magnetism arises in the effective

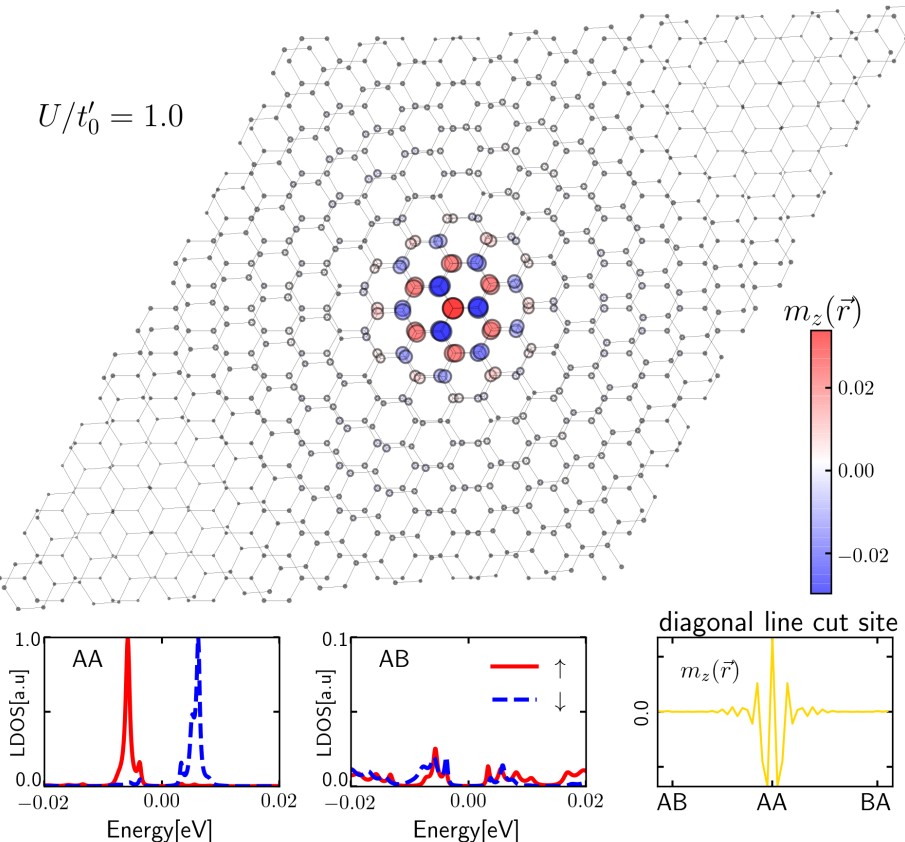

Figure 3: Top panel: MFT result for the spatial magnetization profile of a rescaled twisted bilayer with $\theta_{\text{eff}} = 1.08°$, and the on-site Coulomb interaction $U/t'_0 = 1$. The bottom panels show the local density of states (LDOS) in the AA and AB regions for both spin projections (left two panels), and a diagonal line cut of the local magnetic moment (right panel).

twisted bilayer model at the magic angle for Coulomb interactions $U$ that are an order of magnitude smaller than for decoupled single graphene layers. It should be noted that the $\boldsymbol{q} = \boldsymbol{0}$ magnetic solution considered here only corresponds to a local, but not the global minimum of the energy such that the true critical value of $U_{c1,\text{MFT}}^{1.08°}$ is probably even smaller than the result (8) ($U_{c1,\text{MFT}}^{1.08°} \approx 0.15\, t'_0$ according to the RPA analysis of appendix A).

Figure 2 also includes two examples for larger twist angles $\theta_{\text{eff}} = 1.30°$ and $1.47°$ (green and blue data, respectively). Many of the preceding remarks also apply to these two cases such that we focus on their peculiarities. Remarkably, the case $\theta_{\text{eff}} = 1.30°$ yields an even smaller $U_{c1,\text{MFT}}^{1.30°}/t'_0 \approx 0.21$ than for $\theta_{\text{eff}} = 1.08°$. Actually, while the velocity at the K point only vanishes at the first magic angle $\theta = 1.08°$, the minibands have a very small bandwidth over the entire range until $\theta = 1.30°$. However, when one goes to $\theta_{\text{eff}} = 1.47°$, the critical value of the onsite Coulomb repulsion increases to $U_{c1,\text{MFT}}^{1.48°}/t'_0 \approx 1.0$. This is still significantly smaller than the critical value of a single layer $U_c^{\text{MFT}}/t \approx 2.09$, but clearly larger than in the two other cases, as expected for minibands close to the Fermi level that now have both a finite Fermi velocity and a significant bandwidth.

For a more detailed discussion of the magnetic state found above $U_{c1}$ but before the system becomes completely magnetic, we show in Fig. 3 results for the $\theta_{\text{eff}} = 1.08°$ system and a representative value of the on-site Coulomb interaction $U/t'_0 = 1$. The top panel shows the spatial structure of the magnetization pattern that we find to be localized in the AA stacking

region. Thus, in this region the magnetic state of the twisted bilayer system resembles that of AA stacked bilayer graphene, but at a significantly lower value of $U$ than would be required for the simple AA system to become magnetic. A different perspective of this magnetic pattern is provided by the lower right panel of Fig. 3 that presents a diagonal line cut of the magnetization. The lower left two panels of Fig. 3 show the spin-resolved local density of states (LDOS) in the AA and AB stacking regions. Interestingly, in the AA region one finds two peaks in the LDOS at low energies that are absent in the AB stacking region. The presence of these peaks correlates with the magnetic state, thus rendering scanning tunneling microscopy (STM) experiments a promising candidate for the detection of such a magnetic state.

## 4.2 Dynamical mean-field theory (DMFT)

Even though MFT has been shown to be remarkably successful to qualitatively describe static [42, 43] and dynamic properties [44] in the semi-metallic phase of single-layer graphene, it is known to become quantitatively less accurate for larger values of $U$. For example, the transition to the antiferromagnetic insulator in the nearest-neighbor hopping case is found at $U_c^{\mathrm{MFT}}/t \approx 2.23$ in MFT [45] while more sophisticated and accurate methods place it at a larger $U_c/t \approx 3.8$ [46–49].

DMFT [34] takes local charge fluctuations into account and thus improves the quantitative treatment of the on-site Hubbard interaction. Indeed, already single-site DMFT shifts the estimate of the critical point to the range $U_c^{\mathrm{DMFT}}/t = 3.5, \ldots, 3.7$ [35], *i.e.,* remarkably close to the most accurate estimates [46–49]. Following previous work, we employ here a real-space version of DMFT [36]. DMFT maps the lattice Hamiltonian Eq. (3) onto a set of quantum impurity problems via the local Green's function for site $i$ inside the moiré supercell [34]

$$G_{i\sigma}(z) = \int \mathrm{d}k \left(z\mathbb{I} - \hat{H}_0(\boldsymbol{k}) - \Sigma_\sigma^r(z)\right)^{-1}_{i,i}. \tag{9}$$

Here $\hat{H}_0$ is the single-particle part of Eq. (3). The main approximation is that the local self-energy matrix for spin projection $\sigma$, $\Sigma_\sigma^r(z)$, that plays the role of a dynamical mean field, depends only on frequency $z$, but not on momentum $\boldsymbol{k}$. Eq. (9) can be used to define a collection of $N_c$ single-impurity Anderson models, that we solve here with the numerical renormalization group (NRG) [50–52] and iterate until self-consistency is reached [53, 54]. We refer to Refs. [35, 36] for details on the procedure and just mention two peculiarities for the present case. Firstly, Eq. (9) requires evidently a combination of integration over the moiré Brillouin zone while at the same time solving coupled problems for the $N_c$ atoms inside the moiré supercell. Secondly, even if the band structure of Fig. 1(b), (c) is almost particle-hole symmetric, there is no strict particle-hole symmetry in the present case in contrast to previous work [35, 36]. Consequently, the chemical potential needs to be adjusted appropriately during each iteration in order to ensure half filling. Since the chemical potential enters into Eq. (9) in a non-linear fashion, this renders the numerical problem even more challenging, thus limiting DMFT not only to the rescaled system, but also the number of $U$-values considered.

Figure 4 presents some DMFT results for the rescaled system with $\theta_{\mathrm{eff}} = 1.08°$, *i.e.,* $N_c = 868$. Comparison of the DMFT results for the magnetization versus $U/t_0'$ in Fig. 4 with the MFT results of Fig. 2 shows qualitatively similar behavior. At a technical level, the DMFT results are a bit more noisy. This is due to the logarithmic frequency discretization inherent to NRG [52], to DMFT being generally numerically more expensive, and in particular the difficulty to adjust the chemical potential appropriately. Nevertheless, the main quantitative difference remains that the critical $U_c$ of a single layer is pushed to larger values as compared to simple MFT, and so is the phenomenon of a magnetization arising in the AA stacking region of the twisted bilayer system. Nevertheless, also DMFT clearly detects a magnetization in the

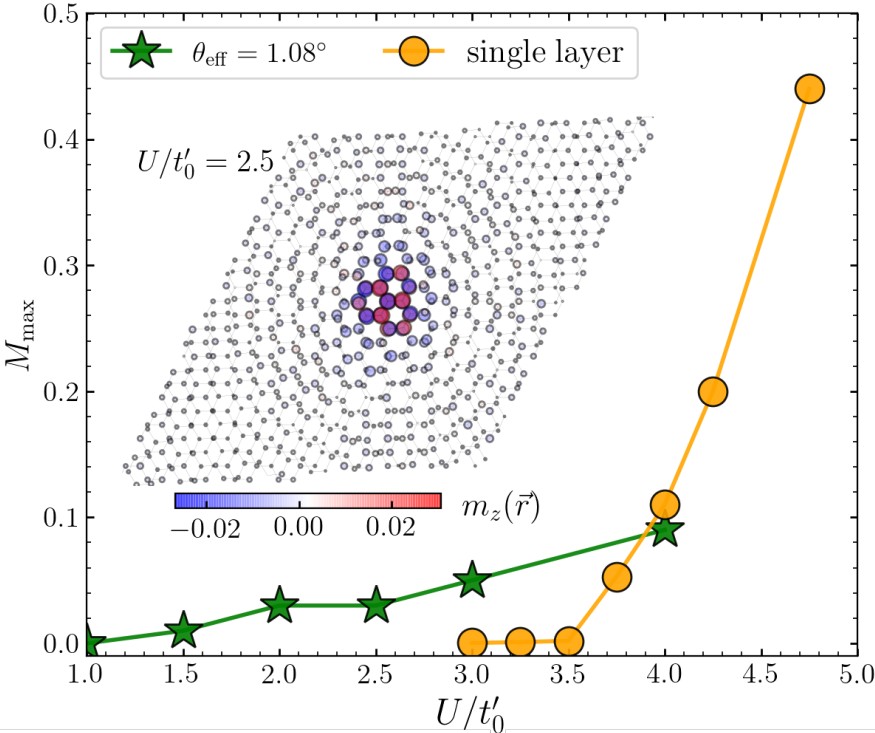

Figure 4: DMFT results for the magnetization versus Hubbard interaction $U/t_0'$ for the rescaled system with $\theta_{\text{eff}} = 1.08°$. For comparison, results for a single graphene layer are also shown. Lines are guides to the eye. The inset shows the spatial magnetization for $U/t_0' = 2.5$.

twisted system for values of the local Coulomb interaction down to $U/t_0' = 1$, amounting to a reduction of the critical value as compared to the single-layer system by at least a factor 3.5 at $\theta_{\text{eff}} = 1.08°$. The inset of Fig. 4 shows an example of the spatial magnetization pattern. This is again very similar to the MFT result shown in the main panel of Fig. 3, just the value of $U/t$ is renormalized to larger values, namely from 1 for the MFT example to 2.5 of the DMFT example. Note that $U/t_0' = 2.5$ would give rise to a bulk magnetic state within MFT while the DMFT result in the inset of Fig. 4 is still clearly localized in the AA stacking region. Overall, DMFT confirms the qualitative conclusions derived from MFT; it just provides a quantitatively more accurate account of the local Coulomb interaction $U$.

## 5 Non-scaled system

We will now present some results for the non-scaled system. The scaling trick has allowed us to apply the quantitatively more accurate DMFT, but the size of the moiré cells of the non-scaled systems will exceed those accessible to DMFT such that we focus on static MFT in the present section. We use the same parameters as in section 4.1 (convergence criterion $10^{-6}$, $9 \times 9$ $\boldsymbol{k}$-grid).

Figure 5 shows MFT results for the total magnetization per moiré cell as a function of $U/t_0$ at rotation angles $\theta = 1.08°$, $\theta = 1.30°$, and $\theta = 1.47°$. The corresponding moiré cells contain $N = 11164$, 7804, and 6076 carbon atoms, respectively. At first sight, the behavior is very similar to that found in the inset of Fig. 2(a) for the rescaled system (the smaller number of data points is due to the significantly enhanced computational effort). In particular, $M_{\text{total}} \lesssim 2$

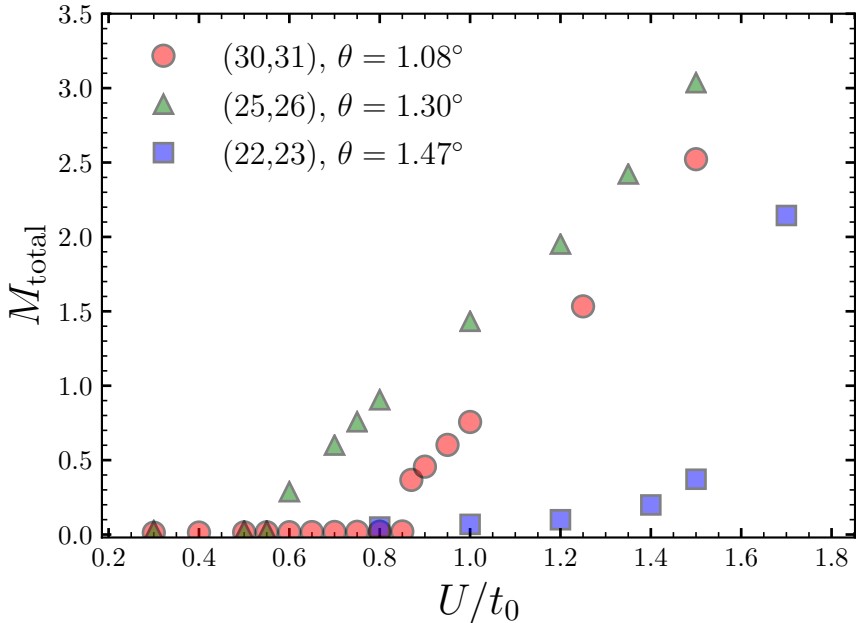

Figure 5: MFT results for the magnetization of the non-scaled twisted bilayer system as a function of $U/t_0$ at rotation angles $\theta = 1.08°$, $\theta = 1.30°$, and $\theta = 1.47°$, respectively.

remains true for most values of $U/t_0$ shown in Fig. 5, in agreement with again the magnetism beging due to the four flat minibands that are closest to the Fermi level.

The key items are the values of the critical Coulomb interaction that one may estimate as $U_{c1,\text{MFT}}^{1.08°}/t_0 \approx 0.85$, $U_{c1,\text{MFT}}^{1.30°}/t_0 \approx 0.55$, and $U_{c1,\text{MFT}}^{1.47°}/t_0 \approx 1$ with a particularly large uncertainty on the last result given the very slow onset of magnetization for $\theta = 1.47°$. According to Ref. [40], in the given normalization, these values should correspond to those found in the rescaled system. This works out more or less for the case $\theta = 1.47°$ where in both cases, the critical $U/t$ ratio is close to 1. However, the values for $U_{c1,\text{MFT}}^{1.08°}$ and $U_{c1,\text{MFT}}^{1.30°}$ in the non-scaled system are bigger than those we might have expected from the rescaled case. Indeed, the order of the discrepancy corresponds to another factor $\Lambda$ such that $U$ scales with $\Lambda^2$ and not just with $\Lambda$. A possible interpretation of this observation is the following: $\Lambda$ actually also appears in the scaling of the linear length [40]. Now the magnetic instability at the angles $\theta = 1.08°$ and $1.30°$ is related to a state localized in the AA region, see, e.g., top panel of Fig. 3. Thus, the area of the relevant spatial region scales with $\Lambda^2$, accordingly the number of contributing local on-site repulsions also scales with $\Lambda^2$ such that $U$ should also scale with $\Lambda^2$ rather than $\Lambda$ in the cases where the physics is controlled by localized states.

In spite of this additional factor, it remains true that $U_{c1,\text{MFT}}^{1.30°}/t_0 \approx 0.55 < U_{c1,\text{MFT}}^{1.08°}/t_0 \approx 0.85$, and that there is still a significant reduction by factors of 4 respectively 3 with respect to the critical value $U_c$ for a single graphene layer. In light of the preceding observations, we suggest that not only the non-interacting bandwidth, but also the size of the moiré cell matter. While both the $\theta = 1.08°$ and $1.30°$ bilayers have a small bandwidth, the moiré cell of the latter is smaller, and this appears to result in a smaller critical value of $U_{c1}$. The size of the moiré cell is smallest for $\theta = 1.47°$ among the three cases studied, but the value of $U_{c1}$ is biggest, most likely because in this case the minibands closest to the Fermi energy are no longer flat. Nevertheless, even in this case one observes emergence of magnetism for values of $U$ that are about a factor 2 smaller than would be needed to render a single layer antiferromagnetic.

To conclude this discussion, let us have a closer look at the spatial structure of the resulting

$\theta = 1.47°$
$U/t_0 = 1.50$

$\theta = 1.08°$
$U/t_0 = 1.0$

Figure 6: The top and bottom panels show the spatial magnetization profile of non-scaled systems at $\theta = 1.47°$ and $\theta = 1.08°$, respectively. The corresponding numbers of atoms in the unit cell are $N_c = 6076$ and $N_c = 11164$.

magnetic states. Figure 6 shows the spatial magnetization profile for non-scaled moiré unit cells with angles $\theta = 1.47°$ and at the first magic angle $\theta = 1.08°$. For illustration purposes, we consider a value of $U$ just above the first critical point $U_{c1}$, *i.e.* $U/t_0 = 1.50$ and $1.00$, respectively. Like for the recaled system shown in the top panel of Fig. 3, we find an anti-ferromagnetic pattern that is localized in the AA region. However, thanks to the improved spatial resolution, we can now observe a clearer separation of the magnetic regions between neighboring cells. For slightly larger $U$, the magnetic region grows, but the structure remains qualitatively similar as in Fig. 6.

# 6 Conclusions and perspectives

We have investigated the onset of magnetism in charge-neutral "magic-angle" twisted bilayer graphene with numerical real-space static and dynamical mean-field approaches. In the rescaled system we found that localized magnetic states appear in the twisted bilayer system for values of the on-site Coulomb repulsion $U$ that are an order of magnitude smaller than those needed to render a single layer magnetic. We then showed that the non-scaled system exhibits qualitatively similar behavior. The reduction is less impressive (up to a factor 4 in the cases investigated), but still remarkable. We note that this finding is consistent with a recent diagrammatic real-space mean-field study [55] that focused on two selected values of $U$.

The rescaling proposed in Ref. [40] actually reproduces the flat minibands close to the Fermi level very well, compare Fig. 1. Our results therefore demonstrate that the band structure is not the only factor that matters. Indeed, the corresponding states are localized in AA stacking regions. This suggests a scaling of the critical $U_c$ with area rather than linear size, as is indeed roughly consistent with our findings for $\theta = 1.08°$ and $1.30°$. A more quantitative analysis would involve computation of the Coulomb matrix elements with respect to the Wannier functions [11,15,56–59] of the rescaled and non-scaled systems, respectively. However, such an analysis goes beyond the scope of the present work.

A side effect of the observation that the spatial extent of the localized states also matters is that smaller unit cells favor magnetism over bigger ones. Indeed, we find onset of magnetism for $\theta = 1.30°$ for smaller values of $U_c$ than for the first magic angle $\theta = 1.08°$. The system with $\theta = 1.47°$ has an even smaller unit cell than that with $\theta = 1.30°$, but at this larger angle there is no longer any really flat band close to the Fermi level such that here the value of $U_c$ is found to be larger. A related point is that magic angles are usually defined via a vanishing Fermi velocity [27–31] while in fact it may be more relevant that the entire minibands are narrow. Indeed, the latter criterion is satisfied over the entire range $\theta = 1.08 \dots 1.30°$ such that the smaller unit cell can then give rise to a lower $U_c$ at the upper boundary of this range of angles.

It should be noted that in our mean-field investigations we have focussed on antiferromagnetic solutions that are periodic over moiré cells. However, the RPA analysis of appendix A suggests that there are other competing instabilities, and indeed the mean-field self-consistency loop sometimes converges to other solutions. In particular, the true lowest-energy state might be modulated in real space and exhibit an internal ferromagnetic structure, like in the case of an electric bias between the two layers [40]. Should this indeed be the case, this can only further reduce the value of the $U_c$ for the onset of magnetism such that our estimates are in fact upper bounds. The main conclusion that twisting leads to a significant reduction of the critical $U_c$ for the appearance of magnetism is thus unaffected by the assumptions on the nature of the ground state.

Another point to note is that we find a stronger reduction of the critical interaction $U_c$ at charge neutrality than a previous RPA investigation [37]. This difference can be traced to a different tight-binding model at the starting point. Indeed, the authors of Ref. [37] have implemented the corrugation of Ref. [56] that takes a modulation of the distance between the two layers in different stacking regions into account. However, other factors may also be relevant in an experiment such as strain when the bilayer is deposited on a substrate. In the same spirit, Coulomb interactions should actually be long-range [60], at least for free-standing bilayers, since atomically thin layers cannot screen the Coulomb repulsion between electrons. Still, screening will depend on the actual substrate and may thus depend on the exact experimental conditions. Even other factors such as spin-orbit interactions that are sufficiently weak to be usually negligible in graphene may matter in the present situation given the significant reduction of the kinetic energy scale in the twisted bilayer system. Thus, which of several competing

instabilities finally wins in an experimental realization may depend on a number of factors; here we have simply demonstrated that a magnetic instability (possibly an antiferromagnetic one) is one of the competitors in charge-neutral twisted bilayer graphene.

The macroscopic magnetization of a ferromagnetic state can be detected, e.g., via the Hall effect [32]. Antiferromagnetic or almost ferromagnetic, but modulated spiral states are more difficult to detect experimentally since they do not give rise to a macroscopic moment. In bulk systems, one would resort to (neutron) scattering to detect such states, but in the present nanoscopic setting this may not be feasible. The best option may thus be scanning tunneling spectroscopy (STS) experiments [61] in order to detect the corresponding characteristic features in the local density of states (see lower panels of Fig. 3). In fact, the corresponding signatures might already have been observed in recent STS experiments [62–64]. However, the latter samples are subject to heterostrain [65,66] which also gives rise to a splitting in the electronic density of states. An unambiguous detection of a magnetic state would thus require a detailed investigation of the variation of the tunneling spectrum with the different stacking regions.

Returning to theoretical questions, an alternative approach would be via low-energy continuum models in the spirit of Ref. [30]. One reason why we have rather used the rescaled model [40] here is that, as illustrates Fig. 1, it reproduces the band structure well over a wide range of energies and not just the flat minibands close to the Fermi level. However, in the range of intermediate values of $U/t$ where mainly the flat minibands contribute to the magnetism, effective low-energy models would have the advantage of being more amenable to numerical approaches [14,67–70] such that we suggest the investigation of magnetism by such methods as a topic for further studies.

A further interesting issue that goes beyond questions accessible to low-energy effective models would be the full phase diagram of the twisted bilayer systems up to larger values of $U/t$. Indeed, the results underlying Fig. 2 suggest that there is no single simple transition to a bulk magnetized system, but that this transition actually proceeds via several intermediate states in the region $U/t'_0 \approx 2$. Given that magnetic interactions in the AB and BA stacking regions are geometrically frustrated (compare Fig. 1(a)), even the magnetic state in the Heisenberg limit $U/t \to \infty$ is far from obvious.

# Acknowledgements

This work was supported by the ANR project J2D "Atomically sharp junctions based on stacked 2D materials: new building blocks for the electronics" and the Paris//Seine excellence initiative. R.P. is supported by JSPS, KAKENHI Grant No. JP18K03511. MFT calculations have been performed at the Centre De Calcul (CDC), CY Cergy Paris Université and using HPC resources from GENCI-IDRIS (Grant No. 910784). We thank Y. Costes and B. Mary, CDC, for computing assistance. DMFT computations in this work have been done using the facilities of the Supercomputer Center at the Institute for Solid State Physics, University of Tokyo.

# A    Noninteracting susceptibility of the rescaled system

In this appendix, we provide an RPA analysis of the noninteracting susceptibility that is similar in spirit to Ref. [37]. However, here we focus on the rescaled system with $\theta_{\text{eff}} = 1.08°$.

We adopt the multiorbital RPA approach to study the instability of the paramagnetic state [71,72].

The multiorbital spin susceptibilities tensor can be expressed in terms of the Matsubara

spin-spin correlation function:

$$\left[\chi(\boldsymbol{q},\omega)\right]_{st} = \frac{1}{3}\int\limits_0^\beta \mathrm{d}\tau\, \mathrm{e}^{i\omega\tau}\left\langle T_\tau \hat{S}_s(\boldsymbol{q},\tau)\hat{S}_t(-\boldsymbol{q},0)\right\rangle, \tag{10}$$

with the Matsubara frequency $\omega$, the imaginary time $\tau$ and spin operators $\hat{\boldsymbol{S}}$ at orbitals $s$, $t$. The noninteracting (zero-order) susceptibility is just a simple bubble diagram involving two Green's functions. Using the spectral representation of the Green's functions, this can be expressed as

$$\left[\chi_0(\boldsymbol{q},i\omega)\right]_{st} = -\frac{1}{N_c}\sum_{\boldsymbol{k}}\sum_{\mu,\nu}\frac{a_\mu^s(\boldsymbol{k})a_\mu^{t*}(\boldsymbol{k})a_\nu^s(\boldsymbol{k}+\boldsymbol{q})a_\nu^{t*}(\boldsymbol{k}+\boldsymbol{q})}{i\omega+E_\nu(\boldsymbol{k}+\boldsymbol{q})-E_\mu(\boldsymbol{k})}\left[n_F(E_\nu(\boldsymbol{k}+\boldsymbol{q}))-n_F(E_\mu(\boldsymbol{k}))\right], \tag{11}$$

where $\mu$, $\nu$ are band indices, $a_\mu^s(\boldsymbol{k})$ and $E_\mu(\boldsymbol{k})$ are the $\mu$-th eigenvalue and eigenvector of the noninteracting Hamiltonian, respectively, and $n_F$ is the Fermi-Dirac distribution function.

The Coulomb interaction can then be included at the mean-field level and one arrives at a so-called "RPA" (or "Stoner", see, e.g. Refs. [26,73]) formula for the interacting susceptibility

$$\chi(\boldsymbol{q},i\omega) = \frac{\chi_0(\boldsymbol{q},i\omega)}{\mathbb{I}-\chi_0(\boldsymbol{q},i\omega)U}, \tag{12}$$

where in the paramagnetic state we can use $\chi_0$ according to Eq. (11). According to Eq. (12), the static susceptibility $\chi(\boldsymbol{q},i\omega=0)$ diverges whenever $U$ equals one of the eigenvalues of the tensor $\chi_0(\boldsymbol{q},i\omega=0)^{-1}$. One can use this identity to determine the mean-field critical $U_c$, and indeed, the critical $U_c$ of an infinite graphene sheet was originally determined in this manner [45]. The value of $\boldsymbol{q}$ and the corresponding eigenvector yield information about the expected magnetic state for $U > U_c$.

The tensor of Eq. (11) is symmetric, but computing all $N_c^2$ entries for a fixed $\boldsymbol{q}$ is time-consuming since each of them involves a sum over reciprocal space and two sums over all energy levels. In order to keep the CPU time manageable, we have limited the sum $\sum_{\mu,\nu}$ to states that are close to the Fermi energy. The latter approximation is physically justified since the ground-state ordering should be dominated by the quasi-flat bands close to the Fermi energy. A similar approximation to the Matsubara sums has also been used in Ref. [37] except that we use here a more radical sharp cutoff. Nevertheless, we have checked that taking the 50 to 100 states closest to the Fermi energy into account is sufficient to yield no visible truncation effects; we used 200 states to be on the safe side. Since we use a finite grid for the integration over the moiré Brillouin zone, the sum Eq. (11) consists strictly speaking of a finite number of poles. In order to smooth these out, we introduce a broadening parameter and evaluate $\mathrm{Re}\chi_0(\boldsymbol{q},i\omega=i\eta)$ such that we obtain a Lorentzian broadening of width $\eta$ at $i\omega=0$. Apart from the truncation in energy space, the momentum grid, and the broadening parameter $\eta$, the result for $\chi_0(\boldsymbol{q},i\omega=0)$ also depends on temperature $T$. $T=10^{-8}\,t$ seems to be sufficiently low to ensure ground-state physics. However, there is a delicate balance between broadening parameter $\eta$ and the grid in reciprocal space. If $\eta$ is too large, it will smear out any peaks and thus reduce the values of $\chi(\boldsymbol{q},i\omega=0)$. On the other hand, for a too small value of $\eta$, the momentum discretization will become visible. We found that the combination $\eta=5\cdot10^{-5}\,t$ and a uniform $9\times9$ grid of points $(k_x,k_y)$ yield a good compromise such that we will present results for these parameters here.

Figure 7(a) shows the distribution of the leading eigenvalue of the static susceptibility tensor $\chi_0(\boldsymbol{q},i\omega=0)$ in the moiré Brillouin zone. In contrast to single layers and AA-stacked bilayer graphene that prefer a single type of ordering at $\boldsymbol{q}=\boldsymbol{0}$, in the present case the maximal

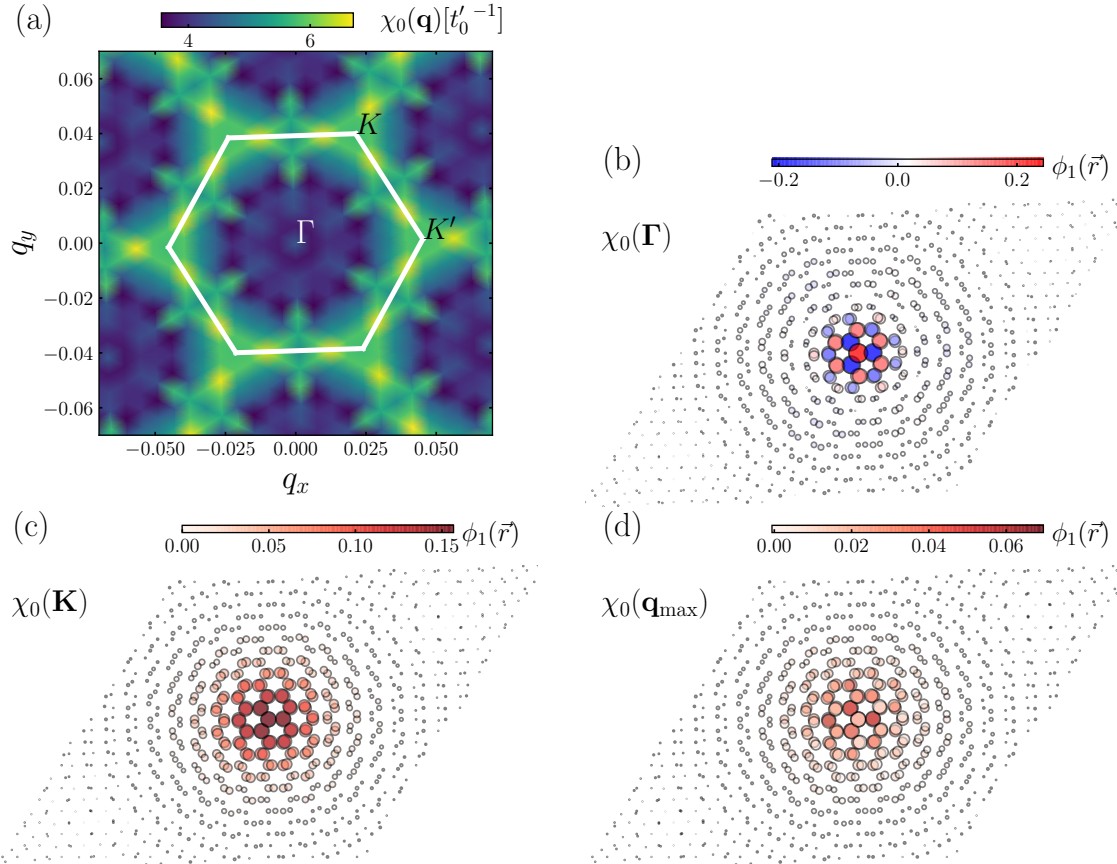

Figure 7: (a) Distribution of the largest eigenvalue $\chi_0(\boldsymbol{q}, 0)$ of the susceptibility tensor for the rescaled system with $\theta_{\text{eff}} = 1.08°$. The white hexagon denotes the first Brillouin zone. Panels (b), (c), and (d) show the spatial profile of the largest eigenvector of the static susceptibility tensor for $\boldsymbol{q} = \Gamma$, K, and $\boldsymbol{q}_{\text{max}}$, respectively. We used $\eta = 5 \cdot 10^{-5}\, t$, a uniform $9 \times 9$ grid to evaluate the sum $\sum_{\boldsymbol{k}}$, and a total of 200 states around the Fermi level for each $\mu$ and $\nu$.

eigenvalue of $\chi_0(\boldsymbol{q}, i\omega = 0)$ is rather flat in reciprocal space. The global maximum is neither at $\boldsymbol{q} = \Gamma$ nor at the two symmetry-related points K and K', but rather at another point $\boldsymbol{q}_{\text{max}}$ at the boundary of the first Brillouin zone. The values are $\max \chi_0(\boldsymbol{q}, i\omega = 0) = 4.35378/t_0'$, $4.99189/t_0'$, and $6.61892/t_0'$, for $\boldsymbol{q} = \Gamma$, K, and $\boldsymbol{q}_{\text{max}}$, respectively. According to the discussion around Eq. (12), this predicts a critical value $U_c = 0.229686\, t_0'$ for a $\boldsymbol{q} = \boldsymbol{0}$ state and globally $U_c = 0.151082\, t_0'$, but for a state with a spatial modulation with a wave vector $\boldsymbol{q}_{\text{max}}$ over moiré cells.

Panels (b)–(d) of Fig. 7 show the corresponding eigenvectors of the susceptibility tensor. At the $\Gamma$ point (panel (b)), one observes a staggered sign change between nearest neighbors with the maxima located in the AA region. This corresponds to the periodic antiferromagnetic state that we have investigated in the main text. The analogous result for the eigenvector at the K (K') point is shown in Fig. 7(c). Here we find a ferromagnetic solution in each moiré cell with the maximum again in the AA region, but the corresponding value of $\boldsymbol{q}$ implies that the corresponding state should be accompanied by a tripling of the unit cell in real space. Finally, Fig. 7(d) shows the eigenvector at $\boldsymbol{q}_{\text{max}}$. The local structure inside a moiré cell is still ferromagnetic, but exhibits a stronger internal modulation. Furthermore, the corresponding MFT solution should be modulated with a wave vector $\boldsymbol{q}_{\text{max}}$ in real space. Examination of further values of $\boldsymbol{q}$ reveals an antiferromagnetic internal structure close to the $\Gamma$ point while

the ferromagnetic internal arrangement is predominant in other regions of the Brillouin zone.

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
