# Peer review of "Magnetism of magic-angle twisted bilayer graphene"

_SciPost Physics, doi:SciPost Phys. 11, 083 (2021)_

## Round 1 · Referee Report · Anonymous (Referee 1) · 2021-6-16

Strengths

Dynamical mean field theory analysis and top the static mean field theory

Weaknesses

Introduction of the rescaled system instead of real large moire cells.

Report

Starting from graphene, a single layer form of the graphite, two dimensional layered materials have drawn great attention in recent years, probably is the most studied systems in the last couple of years within the condensed matter and materials science community after the synthesis of graphene by mechanical exfoliation method in 2004. When the superconductivity observed in twisted bilayer system in 2018, it attracted a lot of attention from scientific community. More than 30 research groups have been studying twisted 2D layered systems since then. This is due the fact that their unusual properties of these simple systems still enrich our understanding of some fundamental physics, even might lead to some interesting applications.

This is a simple but quite interesting study; it provides a discussion about the magnetism of twisted bilayer graphene system around the magic angle of 1.08 degrees. Since, the system size is very large (Nc=11164) at this first magic angle of 1.08 degree, first rescaled system, with twist angle of 3.89 degrees and Nc=868, was introduced and discussed , so that lead a reasonable computational cost and time. Once, this established, then magnetic state of the moire system were investigated by static mean field and dynamic mean field theory based on tight binding Hamiltonian.

In conclusion, I would recommend publishing this manuscript on SciPost.

Requested changes

On page 4, below eq (4), order of intralayer and interlayer should be other way around.

  • validity: high
  • significance: high
  • originality: good
  • clarity: high
  • formatting: excellent
  • grammar: excellent

Author:  Andreas Honecker  on 2021-08-18  [id 1679]

(in reply to Report 1 on 2021-06-16)
Category:
answer to question

We would like to thank the Referee for the kind and detailed appreciation of our work.

Concerning the issue of rescaling of interlayer and intralayer hopping in the sentence following Eq. (4): Ref. [40] indeed proposes to rescale the hoppings in the individual layers while keeping the hopping between the layers fixed. This has the disadvantage of shifting the instability towards antiferromagnetism of individual layers to smaller values of $U$, thus rendering a comparison of rescaled and non-scaled systems at larger values of $U$ difficult. On the other other hand, reproducing not only the low-energy but also the high-energy physics quantitatively would clearly be asking too much for an effective model.

---

## Round 1 · Referee Report · Anonymous (Referee 2) · 2021-7-20

Strengths

1) Original results regarding magnetic properties of twisted bilayer graphene systems. 2) Meanfield Hubbard calculations are complemented with dynamical MF and RPA analysis. 3) Text mostly well written and clear.

Weaknesses

1) Uncertain rescaling of U makes the comparison to single layer difficult.

Report

In this work, magnetic properties of charge-neutral twisted bilayer graphene systems are investigated within Hubbard model using mean-field and dynamical mean-field appraches. In particular, the authors find that, for systems close to magic angles, the critical Hubbard U value to trigger antiferromagnetism insulator phase is significantly lower than for normal bilayer and single-layer graphene.

Most of the results are obtained for "rescaled systems" to reduce computational cost. Some of the results are checked by comparing to non-scaled system results and shown to be consistent. However it is found that rescaled system leads to an overestimated critical U.

I find that this is an interesting, well written and original work on the topical subject of twisted bilayer graphene systems. I can recommend publication of the manuscript in SciPost provided that the authors clarifies following issues:

1) The issue of rescaling of U parameter should be (briefly) introduced earlier in the manuscript since it raises questions about the interpretation of results in Figs.2: should we compare the values of U to single layer results in units of t0' or eV (I tend to think in terms of eV)? Without knowing that U should be rescaled it leads to confusion.

2) In Fig.2b: it seems that the authors plotted the (non-twisted) bilayer system result instead of single layer, since the critical U value seems to be U/t=2.15. Please check the caption and the related text on page 7. Also, are the (non-twisted) bilayer system results for AA or AB?

3) Page 9, first paragraph: I do not understand the sentence starting with "Indeed, already single-site DMFT ...": Why does the right-hand side of the equation has several values?

4) Although the authors find that antiferromagnetism is triggered at lower U values, Fig.2 also shows that a clear increase in magnetization occurs near U/t0' = 2.15. However, this effect does not seem to be present in the DMFT results (Fig.4) which is surprising to me. Can the authors comment on that?

Requested changes

As pointed out in the report section I suggest that the authors should:

1) Add a brief discussion on rescaling of U earlier in the manuscript. 2) Use the right results for single layer system in Fig.2b. 3) Correct/clarify the equation in the first paragraph on page 9. 4) Discuss the discrepancy between MFT and DMFT around Uc .

  • validity: high
  • significance: high
  • originality: high
  • clarity: high
  • formatting: excellent
  • grammar: excellent

Author:  Andreas Honecker  on 2021-08-18  [id 1680]

(in reply to Report 2 on 2021-07-20)

We would like to thank the Referee for the positive and careful appreciation of our work.

Concerning the suggested changes: 1) We moved the content of the sentence "Note that Ref. [40] suggested that the on-site Coulomb interaction should scale in the same way as the intralayer hopping parameters, $U' = \Lambda\,U$" from the beginning of section 5 to section 3. 2) We would like to thank the Referee for pointing out that our analysis for the single-layer case was sloppy. We have now redone calculations for the case of decoupled layers with the same intralayer hopping as in the bilayer system more carefully, replaced Fig. 2(b), and cleaned up the related discussion. Note that we also needed to correct $U^{\rm MFT}_c /t$ slightly to $\approx2.09$. 3) DMFT yields an effective quantum impurity problem that needs to be solved numerically. This numerical solution is subject to numerical errors that depend on the method used, resulting in an uncertainty for the location of the critical value of $U$. The range that we indicated in the second paragraph of section 4.2 corresponds to this numerical uncertainty window. We have added a "the range" and hope that this helps to clarify this point. 4) We understand that this comment of the Referee refers to the data point at $U/t_0'=4.0$ in Fig. 4. As we have emphasized in the manuscript, DMFT computations for this problem are challenging such that the results are subject to numerical errors. This may be particularly true for the twisted bilayer data point at $U/t_0'=4.0$ that lies in the transition region such that we can only speculate about the interpretation. It might be possible that magnetization is indeed reduced in DMFT while it is enhanced in MFT in this transition region. However, it is also possible that this inverted order is just a numerical error within DMFT (like the MFT outlier at $U/t_0'=2.4$ in Fig. 2(b)). As we already emphasize numerical issues in the manuscript, we prefer not to include such speculation in the manuscript. However, there is one further aspect: we have performed DMFT computations only for a few selected values of $U$ given that they are expensive. We have added a comment to this effect to the manuscript and also a comment to the caption of Fig. 4 that lines are only guides to the eye.

---

## Round 2 · Referee Report · Anonymous (Referee 2) · 2021-8-19

Report

The authors have satisfactorily answered all my questions and made relevant changes in the revised version of the manuscript. Therefore I recommend publication in SciPost without further modifications.

---

## Round 2 · Referee Report · Anonymous (Referee 1) · 2021-8-19

Report

I have read the all reviewer reports and comments, the revised manuscript and the replies to the reviewers’ comments. I found that the authors replied the comments and revised the manuscript accordingly, which is improved signifacantly.

In conclusion, I would recommend publishing this manuscript on the SciPost.

---

## Round 2 · Author Response

See replies to Reports for Submission 2104.10694v1 on 23 April 2021.

---

## Round 2 · List of Changes

1) Content of the sentence "Note that Ref. [40] suggested that the on-site Coulomb interaction should scale in the same way as the intralayer hopping parameters, $U' = \Lambda\,U$" moved from the beginning of section 5 to section 3.
2) Replaced Fig. 2(b), and cleaned up the related discussion of decoupled layers.
3) Added a "the range" to the DMFT critical value of $U$ in the second paragraph of section 4.2.
4) Added "Lines are guides to the eye" to the caption of Fig 4 and "thus limiting DMFT not only to the rescaled system, but also the number of $U$-values considered" at the end of the second paragraph of section 4.2.
5) Fixed aspect ratio of Fig. 1(a) and removed unnecessary reference to moiré length.
6) Outlier specification for Fig. 2(b) corrected to $U/t_0'=2.4$.
7) Updated references.
8) Added two references (new Refs. [18,54]).

---

## Editorial Decision

published